# Cardioprotective Role of Heat Shock Proteins in Atrial Fibrillation: From Mechanism of Action to Therapeutic and Diagnostic Target

**DOI:** 10.3390/ijms22010442

**Published:** 2021-01-05

**Authors:** Stan W. van Wijk, Kennedy S. Ramos, Bianca J. J. M. Brundel

**Affiliations:** 1Department of Physiology, Amsterdam Cardiovascular Sciences, Amsterdam UMC, Vrije Universiteit Amsterdam, 1081 HZ Amsterdam, The Netherlands; k.silvaramos@amsterdamumc.nl (K.S.R.); b.brundel@amsterdamumc.nl (B.J.J.M.B.); 2Erasmus Medical Center, Department of Cardiology, 3015 GD Rotterdam, The Netherlands

**Keywords:** atrial fibrillation, heat shock protein, proteostasis, biomarker, HSPB1

## Abstract

Atrial fibrillation (AF) is the most common age-related cardiac arrhythmia worldwide and is associated with ischemic stroke, heart failure, and substantial morbidity and mortality. Unfortunately, current AF therapy is only moderately effective and does not prevent AF progression from recurrent intermittent episodes (paroxysmal) to persistent and finally permanent AF. It has been recognized that AF persistence is related to the presence of electropathology. Electropathology is defined as structural damage, including degradation of sarcomere structures, in the atrial tissue which, in turn, impairs electrical conduction and subsequently the contractile function of atrial cardiomyocytes. Recent research findings indicate that derailed proteostasis underlies structural damage and, consequently, electrical conduction impairment. A healthy proteostasis is of vital importance for proper function of cells, including cardiomyocytes. Cells respond to a loss of proteostatic control by inducing a heat shock response (HSR), which results in heat shock protein (HSP) expression. Emerging clinical evidence indicates that AF-induced proteostasis derailment is rooted in exhaustion of HSPs. Cardiomyocytes lose defense against structural damage-inducing pathways, which drives progression of AF and induction of HSP expression. In particular, small HSPB1 conserves sarcomere structures by preventing their degradation by proteases, and overexpression of HSPB1 accelerates recovery from structural damage in experimental AF model systems. In this review, we provide an overview of the mechanisms of action of HSPs in preventing AF and discuss the therapeutic potential of HSP-inducing compounds in clinical AF, as well as the potential of HSPs as biomarkers to discriminate between the various stages of AF and recurrence of AF after treatment.

## 1. Introduction

Atrial fibrillation (AF) is the most common age-related cardiac arrhythmia worldwide and is associated with ischemic stroke [1,2], dementia [3], heart failure [4], and substantial morbidity and mortality [5]. Prevalence and incidence of AF increase with age and are higher in men compared with women, although women have increased mortality [6,7]. In 2010, an estimated 8.8 million individuals were registered in the European Union with AF, and due to the rapid aging of the population, this number is expected to be doubled by 2060 [5]. AF was originally described as an “electrical” disease and, consequently, current pharmacological treatments are directed at the electrical refractoriness of the atria. Unfortunately, these treatments are only moderately effective and do not prevent AF progression from recurrent intermittent episodes (paroxysmal) to persistent and finally permanent AF stage. Further, usage of pharmacological therapies is limited by potentially severe and life-threatening side effects [8]. In addition to the moderate efficacy of current drug therapies, the current diagnostic tools available have certain limitations to accurately stage AF, especially in regard to AF progression [9]. Currently, AF can only be diagnosed with a surface electrocardiogram when a patient already suffers from AF. Importantly, the electrocardiogram cannot be used to accurately assess the stage of AF, which is important to develop personalized therapies. Therefore, there is an urgent need to develop mechanism-based therapies and identify biomarkers which may help to stage AF. 

Several studies have dissected various mechanisms underlying AF. AF persistence is related to the presence of electropathology, which is defined as structural damage present in atrial tissue, which impairs electrical conduction and subsequently the contractile function of atrial cardiomyocytes [8,10,11,12]. The degree of electropathology is related to the stage of AF and becomes more severe over time [13]. Despite the efforts of current treatment modalities to change the electrical refractoriness of atrial cardiomyocytes, the therapies do not attenuate structural damage and consequently result in accumulation of damage over time. Structural damage includes loss of sarcomeres and myofibrils (myolysis), disruption of mitochondria, and DNA damage [14,15,16]. Deterioration of the cardiomyocytes explains AF progression and the high number of AF recurrence after treatment [17]. 

Emerging research findings indicate that derailed proteostasis underlies structural damage and electrical conduction impairment. A healthy proteostasis is of vital importance for proper function of cells and organisms [18]. Proteostasis is of particular importance in long-lived postmitotic cardiomyocytes, since they display limited regenerative capacity. Proteostasis involves controlling the concentration, conformation, binding interaction, kinetics, and location of individual proteins [19]. Cells respond to a loss of proteostatic control by inducing a heat shock response (HSR), which results in heat shock protein (HSP) expression. The central regulator of the HSR is the Heat Shock Factor 1 (HSF1). This evolutionarily conserved transcription factor becomes trimerized and phosphorylated upon activation while being transported to the nucleus, where it targets heat-shock-responsive DNA elements (HSEs) in order to upregulate the HSP genes (Figure 1) [20,21]. Whereas, under normal physiological conditions, HSF1 is a monomer and mainly present in the cytosol. HSPs such as HSPA1A and HSPC1 bind to and thereby sequester monomeric HSF1, resulting in suppression of HSF1 transcriptional activity [21]. 

It has been recognized that AF-induced proteostasis derailment and subsequent electropathology is rooted in exhaustion of HSPs. HSPs are chaperones which bind to misfolded proteins such as cytoskeletal and sarcomere protein structures, and thereby protect against structural remodeling in AF [14,22,23,24,25,26]. Previous research findings disclosed that some small HSPs, particularly HSPB1, bind to myofibrils and protect against myofibril degradation in experimental and clinical AF (Figure 1) [14]. Furthermore, the HSP response is temporarily activated in patients with short duration of AF but exhausts when AF persists [14]. Consequently, cardiomyocytes lose defense against structural remodeling, thus leading to the progression of AF. Based on these findings, clinical trials have been conducted to test whether HSP inducers can increase HSP levels in human atrial tissue and blood and represent biomarkers to stage AF. In this review, we provide an overview of the cardioprotective effects of HSPs in AF and elucidate the molecular pathways. Furthermore, we discuss the therapeutic potential of HSP-inducing compounds in clinical AF as well as the potential of HSP as a biomarker to discriminate between the various stages of AF and to predict recurrence of AF after treatment. 

## 2. Exhaustion of HSR Underlies AF

The first discovery of HSPs goes back to 1962. Ritossa described the effect of temperature changes on the puffing patterns of chromosomes isolated from the salivary gland of *Drosophila busckii* [27]. Twelve years later, Tissières et al. described specific mRNA expression linked to the chromosomal puffs to become rapidly transported into the cytosol and translated into polypeptides upon heat shock [28]. This response is nowadays known as the HSR, which results in the expression of HSPs. HSPs are ubiquitously expressed molecular chaperone proteins, which exhibit a high level of conservation across all celled organisms. HSPs are involved in a plethora of processes such as protein trafficking, protein folding, and protein complex assembly and therefore play a central role in proteostasis and healthy function of cells [29]. Moreover, HSPs are particularly induced during physiological stress to maintain protein quality control by either refolding misfolded proteins or channeling them to the ubiquitin proteolysis pathway for degradation in order to combat stress-induced protein misfolding [30]. 

Nowadays, 97 different HSPs have been identified, subdivided into several classes based on their molecular weight: HSPA (HSP70), HSPB (small HSP), HSPC (HSP90), HSPH (HSP110), DNAJ (HSP40), and chaperonin families (HSPD/HSP60, HSPE, and CCT) [31]. Failure to mount an adequate HSR is thought to underlie hypersensitivity to acute proteotoxic stress and has been associated with age-related chronic misfolded protein [12,32]. In human AF, there are indications that exhaustion of the HSR underlies this disease. Two studies report that higher atrial expression levels of HSPB1 relate to short duration of AF and less extensive structural damage. In more persistent AF stages, the HSPB1 levels become exhausted [14,22]. Further, an inverse correlation between the amount of HSPB1 and level of myolysis was found [14]. AF patients with high levels of HSPB1 revealed a low amount of myolysis, whereas low levels of HSPB1 were correlated with a high amount of myolysis in atrial tissue samples. In line, in two other studies, a comparable correlation between HSP exhaustion and AF was found. In these studies, an inverse correlation between HSPA1A levels in atrial tissue and the incidence of postoperative AF in patients undergoing cardiac bypass surgery was found [33,34]. 

Thus, studies indicate that in short-duration AF, the HSR is activated, while it diminishes over time when AF persists. As HSPs conserve proteostasis and thereby underlie cardiomyocyte structure and function, lower levels of HSP may trigger the progression of structural damage, paving the way to longstanding and permanent AF. Therefore, securing HSP levels at an adequate level, for example, by treatment with HSP inducers, may limit the expansion of the AF substrate during paroxysmal and short-term AF. In agreement with this hypothesis, restoration of sinus rhythm in patients with permanent AF after mitral valve surgery is correlated with HSF1 activity and induced HSPB1 levels [35]. These findings emphasize the importance of HSF1 and, subsequently, HSPB1 expression to combat AF. Several studies have been conducted to dissect the molecular mechanism underlying HSR exhaustion in AF and have identified targets for HSR boosting.

## 3. Boosting HSR via the Ras Homolog Gene Family Member A (RhoA) Pathway

Why atrial cardiomyocytes are unable to mount a proper HSR in AF is possibly related to activation of the Ras homolog gene family member A (RhoA). RhoA represents a major stress signaling pathway, which becomes activated during AF progression [36,37]. Pathological RhoA activation has been found to suppress the cardioprotective HSR in HL-1 atrial cardiomyocytes by impairing the binding of HSF1 to HSE in the promotor sequence of the HSP genes (Figure 1) [38]. As such, RhoA activation resulted in the inhibition of HSP expression and hypersensitization of cardiomyocytes to proteotoxic stress. Moreover, pre- or post-treatment with the potent HSP booster geranylgeranylacetone (GGA) could not overcome the HSPA1A-suppressive effect of RhoA activation. Importantly, genetic inhibition of RhoA resulted in a strong induction of the HSR, indicating that inhibition of RhoA is a target to increase the HSR and subsequent HSP levels [38]. The mechanism behind the prohibition of RhoA for HSF1 to bind to the HSE needs further investigation, but this pathway opens novel insights in potential druggable targets within the RhoA pathway, which may increase cardioprotective HSP levels in AF [38]. 

## 4. Pharmacological Boosting of HSR with Compounds

Besides inhibition of the RhoA pathway to induce HSR, other studies have tested pharmacological pretreatment with GGA or genetic overexpression of HSPB1 to maintain proper cardiomyocyte function and structure after tachypacing [14,36,39,40,41,42,43]. From a clinical perspective, it would be highly relevant to pharmacologically halt the existing structural damage and contractile dysfunction in atrial cardiomyocytes. This suggestion has been further explored by pharmacological induction of HSP levels with GGA. Pretreatment of dogs with GGA revealed protection against atrial tachypacing and (acute) ischemia-induced impairment in electrophysiology (shortening action potential duration, L-type Ca^2+^ channel current reduction), contractile function (Ca^2+^ transient and cell shortening loss), and conduction velocity and consequently resulted in attenuation of AF onset and progression [39,41]. Further, protective effects of HSPs were observed in a rabbit model for AF induced by heart failure [42].

Notwithstanding GGA’s protective effects, the poor physicochemical properties of GGA, including its hydrophobic nature and limited solubility, may pose disadvantages to its druggability in AF. The gut mucosal distribution pattern owing to GGA’s hydrophobic character hinders its systemic bioavailability [44,45] and therefore likely requires high dosages to treat AF patients. To overcome these disadvantages, various GGA derivatives with improved physicochemical properties compared with GGA have been synthesized and tested for their ability to induce HSPs in HL-1 cardiomyocytes, such as GGA-31, GGA-59, and GGA-60 [46]. Importantly, these derivatives, which are based on one geranyl group instead of two, revealed improved protection against tachypacing-induced Ca^2+^ transient loss in HL-1 atrial cardiomyocytes. The cardioprotective actions of these HSP inducers were HSPB1 dependent, as suppression of HSPB1 by siRNA precluded protection [46]. In addition, these GGA derivatives also protected against tachypacing-induced heart wall dysfunction in a *Drosophila melanogaster* model for AF [46]. 

Importantly, GGA and GGA derivatives were also tested for their ability to reverse structural damage in tachypaced HL-1 atrial cardiomyocytes [25]. In this AF recovery model, GGA and three GGA derivatives, of which GGA-59 was superior, showed an accelerated restoration from tachypacing-induced Ca^2+^ transient loss. GGA-59 likely confers its protective and recovery effects on contractile dysfunction by enhancing HSF1 hyperphosphorylation with the subsequent induction of cardioprotective HSP expression (Figure 1) [46]. However, how GGA and GGA derivatives prolong hyperphosphorylation needs to be further elucidated. There are indications that RhoA might be involved in this process, as post-translational prenylation with C15 (farnesyl) or C20 (geranylgeranyl) isoprenoids mediates translocation of RhoA to the plasma membrane, which results in activation of the downstream RhoA signaling pathway [47,48,49]. GGA and GGA derivatives may compete with endogenous geranyl groups, which could lead to inhibition of physiological RhoA activation, resulting in enhanced binding of HSF1 to the HSE in the promotor region of HSP genes. Interestingly, GGA derivatives with a farnesyl group did not induce HSP expression, suggesting that the geranyl group is required for HSP induction [46]. Further research should elucidate the role of RhoA in GGA-induced HSP expression in cardiomyocytes.

In addition to GGA-related protective effects, other studies have described activation of HSF1 via the semi-essential amino acid L-glutamine to boost intracellular levels of HSPs (Figure 1) [50,51,52]. Moreover, L-glutamine has been described to be protective against ischemic heart disease and heart failure by induction of HSPs [53,54]. Moreover, a recent study investigated the role of L-glutamine on serum HSP levels in patients with AF. This study found that HSPB1 and HSPA1A in the serum of AF patients dropped after 3 months of daily L-glutamine supplementation. Further, a strong inverse correlation was observed between serum levels of HSPB1 or HSPA1A at baseline and after 3 months of L-glutamine supplementation. Interestingly, metabolites in the carbohydrate, nucleotide, and amino acid synthesis pathway were normalized in AF patients after 3 months of L-glutamine supplementation [55]. These findings indicate that L-glutamine supplementation affects HSP levels in AF patients and normalizes the metabolic pathways, which may attenuate proteostasis derailment and structural damage in cardiomyocytes in clinical AF. 

It has also been shown that the hydroxylamine derivate BPG-15 provides protection against experimental AF by overexpression of endogenous dmHSP23 (homologous to human HSPB1) in a *D. melanogaster* model [43]. In contrast, a mouse model for heart failure and AF showed that the protective effect of BGP-15 was independent of HSPA1 and HSPB1, but attributed the drug’s protective role to the phosphorylation of insulin-like growth factor 1 (IGF-1) receptor [56]. Although it has been described that hydroximic acid derivatives can induce HSPs, the exact molecular mode of action of BGP-15 within AF has not been elucidated yet [57]. Interestingly, a study in rats showed that BGP-15 also acts as an inhibitor of poly(ADP)-ribose polymerase 1 (PARP-1) [58]. Since a previous study revealed a role for DNA-damage-induced PARP1 activation to prevent structural damage in clinical AF, BGP-15 may also protect against AF via PARP-1 inhibition [16]. Whether HSPs are involved in the DNA-damage–PARP1 axis is unknown. 

Although AF results in HSR exhaustion and cardiomyocyte dysfunction, it also may result in cardioembolic stroke. Interestingly, studies showed that induction of HSPA1 with either HSF1 inducer TRC051384 or the HDAC6 inhibitor tubastatin A delayed the development of thrombi in a murine model for thrombus formation [59]. HSPA1 did not affect bleeding in mice, which is in contrast to normal anticoagulants. This is especially relevant considering anticoagulant treatment in patients increases the risk for major bleeding. Bleeding in patients could ultimately halt treatment with anticoagulants, which were administered to prevent cardioembolic stroke [59]. Further, supplementation of peroxisome proliferator-activated receptor (PPAR)-γ pioglitazone was shown to upregulate HSPA1 mRNA and protein levels in atrial tissue of a rat model of AF, which resulted in attenuation of fibrosis and morphological changes [60].

## 5. GGA Induces HSP Levels in Atrial Tissue of Patients Undergoing Open-Heart Surgery

A feature which makes GGA and GGA derivatives interesting as drugs is that they only boost the HSR in cardiomyocytes under mild stress conditions [46]. This indicates that augmentation of the HSR by GGA and its derivatives is confined to stressed cells, avoiding potential side effects due to enhanced HSR in non-stressed cells. GGA is already marketed in various Asian countries, despite the less favorable physicochemical therapeutic profile, and is considered a safe drug. Therefore, GGA itself may already be further explored in clinical AF. A recent study obtained proof of concept for its HSP-inducing effect in human atrial tissue, since it was still unknown whether GGA can induce HSPs in the human heart [26]. Three days of oral GGA treatment (400 mg/day), prior to coronary artery bypass grafting surgery in patients with coronary artery disease, was associated with a significant increase in HSPB1 and HSPA1 expression levels in right and left atrial appendages. Furthermore, increased HSPB1 levels were present at the myofilaments in patients treated with GGA, suggesting a beneficial effect by conservation of the myofilament structures [26]. These findings pave the way for further studies on the role of HSP induction by GGA and/or GGA derivatives to protect against (postoperative) AF. 

## 6. Accelerated Restoration of Structural Damage by HSPB1: Mode of Action

As virtually all AF patients have already developed a level of structural damage at the moment of diagnosis, it is clinically highly relevant to explore whether HSPs can reverse deterioration. Fundamental insights into HSP-induced structural restoration of cardiomyocytes will boost further clinical research to improve the recovery of heart function after AF conversion and may help to prevent AF onset. Experimental and clinical AF research findings indicate that boosting of HSPB1 seems to be crucial for the protective actions of GGA and GGA derivatives, as suppression of HSPB1 precluded protection [39,46]. Therefore, it is of interest to dissect the molecular mode of action of HSPB1 in recovery from AF. Is has been recognized that HSPB1 (co)localizes at the myofilaments; stabilizes the sarcomeric proteins, including alpha-actinin, actin, and myosin; and as such conserves cell structure [14,61,62]. By binding to and stabilizing the sarcomeric proteins, HSPB1 may shield the contractile proteins from AF-induced cleavage by cysteine proteases, including calpain 1 (Figure 1) [43,63,64]. Experiments with post-tachypacing-induced HSP expression reveal that HSPB1 also accelerates recovery. HL-1 cardiomyocytes post-treated with GGA-59 revealed accelerated recovery from tachypacing-induced Ca^2+^ transient loss and restored mRNA and protein levels of (acetylated) α-tubulin, as well as protein levels of cardiac troponin I and troponin T. Similar protective effects were observed after overexpression with recombinant HSPB1 with Ca^2+^ transient loss and tubulin expression; however, this did not affect cardiac troponins [25]. 

Furthermore, post-treatment with GGA-59 enhanced recovery of depolymerized (acetylated) α-tubulin fractions, which coincided with elevated HSPB1 binding. GGA-59 treatment did not change tachypacing-induced calpain activity but did normalize HDAC6 activity [25]. Interestingly, HDAC6 activity was previously found to result in deacetylation of the microtubule network and AF promotion in experimental cardiomyocyte (*Drosophila*) and dog models for AF [65]. These findings point to HSPB1 protecting the microtubule network, possibly by direct binding of HSPB1 to HDAC6 (Figure 1) [66], and suppressing its activity, resulting in prevention of deacetylation, depolymerization, and subsequent degradation of α-tubulin by calpain [25]. Since overexpression of recombinant HSPB1 caused a restoration of post-tachypacing α-tubulin expression, it can be hypothesized that HSPB1 directly influences α-tubulin transcription. Moreover, a previous study showed that HSPB1 interacted with the transcription factor SP1 and thereby regulated gene transcription in experimental cell culture models [67]. However, whether HSPB1 interacts with transcription factors that target tubulin genes remains unknown. Another explanation for the restoration of α-tubulin mRNA expression might be the indirect effect of HSPB1 on the microtubule network, as depolymerized α-tubulin can degrade and destabilize α-tubulin mRNA associated with ribosomes [68,69,70]. It remains to be elucidated whether this autoregulatory mechanism is activated upon tachypacing of the atrial cardiomyocyte. Thus, induction of HSPB1 levels accelerates recovery from tachypacing-induced structural damage and contractile dysfunction in HL-1 cardiomyocytes, indicating that HSP induction is an interesting target to potentially reverse AF-induced remodeling. 

## 7. Potential Protective Role of HSPA in AF 

The HSPA protein family has an essential role in proteostasis, and a protective role in AF has been described. In an experimental study in mice, it was observed that induction of HSPA1 prevents angiotensin-II-induced AF by limiting atrial fibrosis formation [71]. Interestingly, in a dog model, it was shown that supplementation of angiotensin (1–7) protects against tachypacing-induced sustained AF and correlates with a decrease in HSPB1 mRNA and protein levels in the left and right atria [72]. Angiotensin (1–7) likely antagonizes angiotensin II and in turn prevents atrial remodeling, thereby avoiding an HSR [72]. Furthermore, a recent study discovered the interplay between HSPA and a single nucleotide polymorphism (SNP) in nuclear lamins. The SNP mutation p.R399C in LMNA was found in a sporadic case during a whole-exome sequencing of more than 600 patients with “lone” AF. LMNA p.R399C was suggested to impair the interaction of lamin A/C with nuclear pore complex protein (NUP155). Moreover, impaired interaction between lamin A/C and NUP155 was found to prevent the export of HSPA1, and potentially also other HSPs, out of the nucleus (Figure 1) [73]. As such, mutations in nuclear filaments may underlie the development of clinical AF [73,74,75,76,77]. In line, recent experiments performed in DLD-1 colorectal epithelial cells showed not only attenuated HSPA1 import upon depletion of lamin A/C but also an impaired transcription of HSPA1, indicating a key role for mutant-lamin-induced HSPA suppression [78]. As mutations in lamin A/C are well described in individuals diagnosed with AF and are linked to various cardiomyopathies [75,76,79,80,81], impairment of nuclear export of HSPA to the cytosol may represent an important pathway in mutant-lamin-induced AF [34,82].

Although not regulated by HSF1, a cardioprotective role for HSPA5 was previously identified in AF [83]. HSPA5 is located in the endoplasmic reticulum (ER) of the cardiomyocyte and attenuates AF-induced ER stress and autophagy, thereby diminishing structural remodeling (Figure 1) [83]. Interestingly, HSPA5 was found to be a hub–bottleneck gene in a recent meta-analysis of transcriptome data related to AF [84]. HSPA5 was consistently downregulated, which further establishes the protective function of HSPA5 normalization in AF [84].

## 8. HSPs as Biomarkers in AF

Currently, accurate staging of AF lacks sophisticated diagnostic tools, which hampers the selection of patient-tailored therapy [9]. Consequently, there is an urgent need to identify biomarkers which may help to stage AF. Serum and atrial tissue samples of patients with or without AF were enrolled in the HALT&REVERSE trial in order to test the potency of HSPs as a potential biomarker [85]. This trial investigated whether HSP levels in serum discriminate between control and the various stages of AF and predict AF recurrence after treatment. Importantly, serum HSPA1, HSPB1, HSPB7, and HSPD1 levels did not discriminate between the presence or stage of AF, nor did it identify patients at risk of AF recurrence after electrical cardioversion or pulmonary vein ablation [17]. However, follow-up serum HSPB1 levels predict AF recurrence in patients undergoing ablative therapy. Here, serum HSPB1 levels significantly increased at 3, 6, and 12 months post-ablation compared with patients without AF recurrence within 1 year post-ablation [17]. These results suggest that HSPB1 levels may be useful in predicting recurrence of AF after ablative therapy. These results are further supported by other studies showing an association between increased atrial tissue levels of HSPA1 and HSPB1 and restoration of sinus rhythm in patients after mitral valve surgery [86], and that high HSPB1 levels in blood predict sinus rhythm maintenance after catheter ablation in patients with paroxysmal AF [87].

Whether serum HSPB1 levels can also be used to select patients for HSP-inducing therapy remains to be elucidated. As mentioned above, AF patients with high levels of HSPB1 at baseline and who were treated for 3 months with the HSP-inducing compound L-glutamine showed significant reduction in serum HSPB1 levels accompanied with normalization of metabolite levels compared with patients with low HSP levels at baseline. This all indicates that serum HSPB1 levels may guide patient-tailored therapy [55]. In addition, further research should elucidate whether the HSP levels in serum correlate with HSP levels in atrial tissue and the degree of electrical conduction abnormalities and voltage changes. This could also shed light on whether increased HSP levels are a result of AF or precedes AF (recurrence). Studies with longitudinal blood sampling (for HSP measurements) and continuous monitoring of electrical parameters may pinpoint possible associations.

## 9. Conclusions

The current clinical treatment and diagnostic staging of AF require further improvement. Pharmacological boosting of HSP levels, by inhibition of the RhoA pathway and enhancing binding of HSF1 to HSP genes, in atrial tissue may aid in the accelerated recovery from cardiomyocyte damage and, as such, may result in halting of AF. Identified mechanisms of action include prevention of microtubule disruption by HDAC6 and cleavage by calpain and increase in microtubule transcription. Findings also indicate that HSP levels in serum may stage AF and identify patients at risk of AF recurrence after treatment.

## Figures and Tables

**Figure 1 ijms-22-00442-f001:**
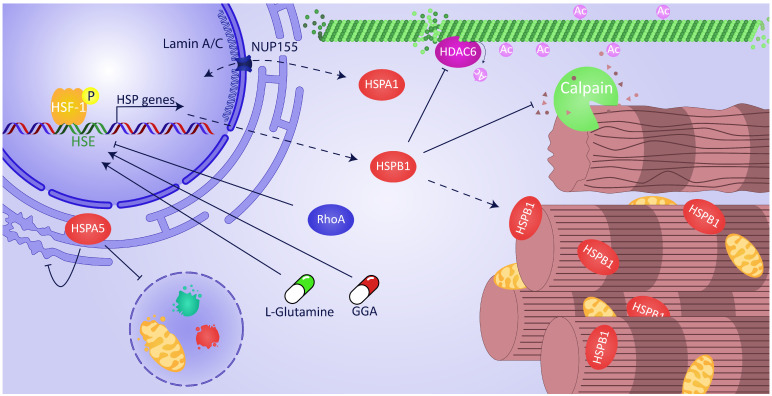
Overview of cardioprotective actions of heat shock protein (HSP) function in atrial fibrillation (AF). Heat Shock Factor 1 (HSF1), the central regulator of the heat shock response (HSR), trimerizes and is phosphorylated upon activation while being transported to the nucleus where it binds to the heat-shock-responsive DNA element (HSE) of HSP genes, including *HSPB1*. The binding of HSF1 to HSE and sequential expression of HSPs is boosted by geranylgeranylacetone (GGA) and L-glutamine but attenuated by active Ras homolog gene family member A (RhoA). Within the cytosol, HSPB1 inhibits HDAC6-induced microtubule degradation by calpain. Furthermore, HSPB1 has a protective function by direct binding to the myofibrils and shielding them from degradation. HSPA5, located in the endoplasmic reticulum (ER), reduces ER stress and inhibits AF-induced autophagy. The export of mRNA HSPA1 and import of protein HSPA1 via NUP155 is directly influenced by mutations in lamin A/C.

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
