# Peer review of "Cardioprotective Role of Heat Shock Proteins in Atrial Fibrillation: From Mechanism of Action to Therapeutic and Diagnostic Target"

_ijms, 2021, doi:10.3390/ijms22010442_

Reviewer 1 Report

I think the review is good.

Many POINTs of STREGHTs are identifiable:

  • Very good quality of graphical abstract
  • Extensive and well-argumented discussion
  • Medium quality of English language (excpet for following considerations - see below)

Nevertheless, I'd like to suggest some modifications to the paper in order to ameliorate it before considering publication by Editors:

  • Starting from minor English revisions. TO BE CHECKED (i.e. : Atrial fibrillation (AF) is the most common age-related persistent cardiac arrhythmia worldwide and is associated with ischemic stroke, heart failure and a substantial morbidity and mortality...)
  • Moreover, you may read the following papers for enlarging/modifying your introduction:
    • Mayo Clin Proc. 2016 Dec;91(12):1778-1810. doi: 10.1016/j.mayocp.2016.08.022. Epub 2016 Nov 5.
    • Stroke. 2016 Mar;47(3):895-900. doi: 10.1161/STROKEAHA.115.012004. Epub 2016 Jan 19.
  • Finally, PLEASE CHECK all references for evaluating of they are all in accordance with authors' rules.

Best regards,

LG

Reviewer 2 Report

The authors provided a review on the cardioprotective role of heat shock proteins in atrial fibrillation. The review is well written and informative. I recommend to accept this manuscript for publication after considering the following suggestions:

General suggestions:
1. The abstract contains a lot of phrases such as ‘Consequently, …‘, ‚In line, …‘In addition, …‘  which should not be used in the abstract. The abstract needs to be specific, clear, and precise. Please change the abstract accordingly.

2. The manuscript should be revised for uniformity (use of abbreviations and, position of spaces)

3. Please have the entire manuscript checked by a native English speaker for grammar and style.

Specific suggestions:
Abstract page 1, line 9: please delete ‘persistent’ as this might be confused with persisting AF

Abstract page 1, line 26: please change ‘experiment’ into ‘experimental’

Introduction page 1, line 33: please delete ‘persistent’ as this might be confused with persisting AF

Introduction page 1, line 42: Please elaborate on the following statement ‘In addition to the moderate efficacy of current drug therapies, there are no diagnostic tools available to accurately stage the AF [6].’ It is hard to say that there are ‘no’ diagnostic tools available. It might be better to say that current diagnostic tools have certain limitations to accurately stage AF especially in regard to AF progression.

Figure 1 page 3, line 90: please change ‘heat shock protein’ into ‘HSP’

Figure 1 page 3, line 92: please change ‘hsp’ into ‘HSP’ (in capital letters and in italics)

Discussion: page 3, line 101: Is it ‘Exhaustion of heat shock stress response underlies AF’ or rather ‘Exhaustion of the HSR underlies AF’ as mentioned in line 120?

Discussion: page 3, line 121: please change ‘Two studies report’ into ‘Two studies reported’

Discussion: page 3, line 124: please delete the addition space between ‘myolysis’ and ‘was’

Discussion: page 4, line 130: please change ‘the HSP response gets activated’ into ‘the HSR is activated’

Discussion: page 4, line 141: please change ‘heat shock response’ into ‘HSR’

Discussion: page 4, line 147: please change ‘hsp’ into ‘HSP’ (in capital letters and in italics)

Discussion: page 4, line 156: Is ‘Pharmacological boosting of heat shock response with compounds` a heading? If so, please put it in Italics.

Discussion: page 5, line 201: please change ‘by inducting of HSPs’ into ‘by induction of HSPs’

Discussion: page 5, line 201: please change ‘described to protect against’ into ‘described to be protective against’

Discussion: page 6, line 253: do you mean ‘exploitation’ or rather ‘exploration’?

Discussion: page 7, line 322: please rephrase ‘Currently, no diagnostics are available to accurately stage AF’ into ‘Currently, accurate staging of AF lacks sophisticated diagnostic tools

Conclusions, page 8, line 353: please change ‘The current clinical treatment and diagnostic staging of AF is inaccurate’ into ‘The current clinical treatment and diagnostic staging of AF requires further improvement’.

Conclusions, page 8, line 355: please change ‘hsp’ into ‘HSP’ (in capital letters and in italics)

Thank you!
